# Bone Tissue Engineering Scaffold Optimisation through Modification of Chitosan/Ceramic Composition

Keran Zhou [ID], Farah Alwani Azaman [ID], Zhi Cao, Margaret Brennan Fournet and Declan M. Devine *[ID]

PRISM Research Institute, Technological University of the Shannon, Midlands Midwest, Athlone Main Campus, N37 HD68 Athlone, Ireland; a00258808@student.ait.ie (K.Z.)
* Correspondence: ddevine@tus.ie

**Abstract:** A large bone defect is defined as a defect that exceeds the regenerative capacity of the bone. Nowadays, autologous bone grafting is still the gold standard treatment. In this study, a hybrid bone tissue engineering scaffold (BTE) was designed with biocompatibility, biodegradability and adequate mechanical strength as the primary objectives. Chitosan (CS) is a biocompatible and biodegradable polymer that can be used in a wide range of applications in bone tissue engineering. Hydroxyapatite (HAp) and fluorapatite (FAp) have the potential to improve the mechanical properties of CS. In the present work, different volumes of acetic acid (AA) and different ratios of HAp and FAp scaffolds were prepared and UV cross-linked to form a 3D structure. The properties of the scaffolds were characterised by scanning electron microscopy (SEM), Fourier transform infrared (FTIR) spectroscopy, swelling studies and compression testing. The cytotoxicity result was obtained by the MTT assay. The degradation rate was tested by weight loss after the scaffold was immersed in SBF. The results showed that a crosslinked structure was formed and that bonding occurred between different materials within the scaffold. Additionally, the scaffolds not only provided sufficient mechanical strength but were also cytocompatibility, depending on their composition. The scaffolds were degraded gradually within a 6-to-8-week testing period, which closely matches bone regeneration rates, indicating their potential in the BTE field.

**Keywords:** bone tissue engineering; ceramic; chitosan; scaffold



## 1. Introduction

Bone is an indispensable and multifunctional organ. The primary function of the adult skeleton is to bear loads and assist in movement. Bone possesses the remarkable ability to heal itself if the defect does not exceed the healing capacity of the individual bone, i.e., if the defect is greater than the critical size [1]. A critical size defect (CSD) is characterised as a defect with a minimum length that cannot heal spontaneously, leading to a non-union [2]. Typically, critical size defects are generally considered larger than 1.5 to 2 times the diameter of the long bone diaphysis, but they vary according to the host and the bone [3]. Globally, more than two million bone graft procedures take place annually, and the bone grafting rate is estimated to increase by 13% each year [4]. Up until now, large bone defects have been a significant challenge in clinical practice. Autologous bone grafts (autografts) still represent the global gold standard treatment for large bone defects, but several factors limit their application (e.g., possible donor site morbidity, long surgery time, limited tissue source and residual pain) [5]. Allografts are derived from living human donors or cadavers [6]. The main advantages of allografts are the availability of the desired graft size with no donor site morbidities and of the large quantities of source tissue [7,8]. However, there is a risk of immunological rejection and graft rejection, as well as high costs [7]. Xenografts are natural-based bone substitute materials derived from animals, commonly coral, porcine and bovine sources [9]. Xenografts have significant drawbacks, such as possible zoonotic disease transmission and transplant rejection. There are also

ethical and religious concerns about using xenografts [6,10]. Bone tissue engineering (BTE) has been widely investigated for use in the bone regeneration process. Considering the significant limitations of traditional therapeutic approaches, BTE has been demonstrated to be one of the most effective treatments for bone defects [11]. The ideal bone graft substitute should have several properties, including osteoinductive, osteoconductive, high porosity and high mechanical strength [12]. A sufficient porosity with suitably sized pores and interconnections between the pores provides an environment to promote cell infiltration, migration, vascularisation, nutrient and oxygen transport and the removal of waste materials while being able to withstand external loading stresses [13–15]. It should also promote the differentiation of progenitor cells into osteoblasts, support bone growth and facilitate bone fusion to form new bone tissue [16]. The biopolymer/bioceramic scaffold has attracted increasing attention in bone regeneration because it might combine the good processability of biopolymers with the excellent mechanical strength of bioceramics while eliminating the risk of brittle fracturing of the scaffold.

Chitosan (CS) is a commercially available cationic, biocompatible and biodegradable biopolymer that has D-glucosamine units and N-acetyl-D-glucosamine units linked by β (1 → 4) [17]. CS is commonly derived from the partial deacetylation of chitin, which is a natural structural polysaccharide extracted from crustaceans, cephalopods and fungi [18]. The procedure of deacetylation exposes the positively charged amine groups of the glucosamine units, which increase the solubility of the polymer [19,20]. CS is an effective material for repairing bone with a structure that resembles the glycosaminoglycans found in the ECM of bone and cartilage [21,22]. CS has desirable characteristics, such as high osteoinductivity, osteointegration and biodegradability, which promote cellular attachment, proliferation and viability. These characteristics make it a good candidate for BTE [9,19]. Regardless of the manufacturing process, the load-bearing capacity of pure CS scaffolds is not satisfactory in most environments. Therefore, to overcome the limitations of CS, it is commonly used in combination with other materials, such as natural polymers, synthetic polymers and bioceramics, to enhance its mechanical properties [23,24].

Hydroxyapatite (HAp) is a major inorganic component of the bone matrix and has favourable biocompatibility [25]. HAp is one of the most frequently utilised bone graft materials [11,26]. Not only is it a microporous material, which offers a good environment for cell adhesion, migration and proliferation, but it also exhibits excellent bioactivity with osteoconductivity and osteoinductivity in biological systems. Owing to its outstanding biocompatibility, HAp can interact with natural tissues without causing significant inflammatory reactions [8].

The limitations of using a HAp scaffold for bone regeneration include poor mechanical stability, difficulty in shaping and fragility. Fluorapatite (FAp) has high chemical and structural stability, which results from the replacement of the hydroxide ($OH^-$) groups in HAp with fluorine ions ($F^-$). Over the past several years, it has been found that $F^-$ released from FAp scaffolds contribute significantly to the stimulation of the attachment, proliferation and differentiation of osteoblast cells [21]. Furthermore, in 2004, Bhadang and Gross [27] showed that FAp integrates well with bone tissue. As such, FAp scaffolds are attractive candidates for bone regeneration applications.

Hence, the aim of this study is to investigate how the composition of the scaffold affects its swelling, degradation and cytotoxic properties. It is hypothesised that the incorporation of FAp will increase the solubility of the scaffold while maintaining favourable swelling and mechanical testing results.

## 2. Materials and Methods

High molecular weight (HW) CS (MW: 310,000–375,000 Da), sodium fluoride, phosphoric acid and acetic acid (AA) (cat.#8187551000) were purchased from Sigma-Aldrich (Wicklow, Ireland). Poly (ethylene glycol) dimethacrylate (PEGDMA) (MW 600 g/mol) was supplied by PolySciences Inc. (Polysciences Europe GmbH, Hirschberg an der Bergstrasse, Germany). Sodium bicarbonate ($NaHCO_3$) and potassium phosphate dibasic trihydrate

($K_2HPO_4 \cdot 3H_2O$) used in this study were supplied by Fisher Scientific (UK Ltd., Loughborough, UK) and benzophenone (99%) ($C_6H_5COC_6H_5$) was purchased from Alfa Aesar (Thermo Fisher Scientific, Kandel, GmbH, Germany).

### 2.1. Synthesising Fluorapatite (FAp)

The process of synthesis involved the steps outlined below.

1. An amount of 0.43 g of sodium fluoride (NaF) powder was added to 10 mL of 10 mol/L phosphoric acid and mixed using magnificent stirrer;
2. Then, 5 g of hydroxyapatite was added to the solution and reacted for few minutes;
3. The powder was mixed in mortar, producing FAp ($Ca_5(PO_4)_3F$) through the following reaction.

$$NaF + H_3PO_4 \rightarrow NaH_2PO_4 \tag{1}$$

$$Ca_5(PO_4)_3OH + 5HF \rightarrow Ca_5(PO_4)_3F + 3H_2O, \tag{2}$$

### 2.2. Scaffold Preparation

For scaffold preparation, different yields of CS pastes were prepared by varying volumes of 1 vol% acetic acid (AA) solution added to CS powder, as outlined in Table 1.

Different ratios of HAp and FAp were mixed with CS pastes. Photopolymerisation of CS/HAp/FAp scaffolds was performed using benzophenone (BP) as a photoinitiator in a UV chamber. The composite paste was placed into a silicone mould with a 25 mm diameter and 2 mm depth and cured for 40 mins. All scaffolds were turned over mid-curing during the process.

**Table 1.** Scaffold formations table.

| Scaffold ID | HAp/(%wt) | FAp/(%wt) | Volume of AA/(mL) |
|---|---|---|---|
| 12.5 $CS_H/HAp_{1.5}/FAp_0$ | 1.5 | 0 | 12.5 |
| 12.5 $CS_H/HAp_{0.7.5}/FAp_{0.75}$ | 0 | 1.5 | 12.5 |
| 12.5 $CS_H/HAp_0/FAp_{1.5}$ | 0.75 | 0.75 | 12.5 |
| 20 $CS_H/HAp_{1.5}/FAp_0$ | 0 | 1.5 | 20 |
| 20 $CS_H/HAp_{0.7.5}/FAp_{0.75}$ | 0.75 | 0.75 | 20 |
| 20 $CS_H/HAp_0/FAp_{1.5}$ | 0 | 1.5 | 30 |
| 30 $CS_H/HAp_{1.5}/FAp_0$ | 1.5 | 0 | 30 |
| 30 $CS_H/HAp_{0.7.5}/FAp_{0.75}$ | 0.75 | 0.75 | 30 |
| 30 $CS_H/HAp_0/FAp_{1.5}$ | 0 | 1.5 | 30 |

CS: chitosan; HAp: hydroxyapatite; FAp: fluorapatite; AA: 1% acetic acid.

### 2.3. Morphological SEM Study

Scanning electron microscopy (SEM) was carried out by Mira XMU SEM (TESCAN Brno, Czech Republic) to visualise and evaluate the surface morphology of the scaffold and distribution of the ceramic. The scaffold was placed onto an aluminium stub using double-sided carbon tape. Scaffolds were sputter-coated with gold using Baltec SCD 005 for 110 s at 0.1 mbar vacuum before testing.

### 2.4. Swelling Study

The swelling characteristics of scaffolds (Table 1; *n* = 3) were assessed by drying scaffolds to equilibrium in a vacuum oven (SalvisLab Vacucenter, Rotkreuz, Switzerland) at 37 °C, 70 mbar for 48 h to obtain the scaffold dry weight ($W_i$). The scaffolds were submerged in pH 7.4 PBS at room temperature for 48 h until they reached swollen equilibrium. The scaffolds were weighed ($W_s$). The equilibrium water content (EWC), % swelling, water uptake ($W_u$) and gel fractions (GFs) were calculated from these data using the following equations:

$$EWC = \frac{W_s - W_i}{W_S} * 100 \tag{3}$$

$$W_u = \frac{W_s - W_i}{W_i} * 100 \tag{4}$$

$$\%\text{swelling} = \frac{W_s}{W_i} * 100 \tag{5}$$

To calculate the GFs of the freshly prepared scaffold, the scaffold was cured and then dried to equilibrium ($W_i$). Then the dried scaffold was immersed in a 1% acetic acid solution for 48 h and dried again ($W_f$). This procedure can analyse the crosslinking reaction efficacy.

$$\text{Gel fractions} = \frac{W_f}{W_i} * 100 \tag{6}$$

### 2.5. Characterisation of Scaffold (Fourier Transform Infrared)

Fourier transform infrared (FTIR) spectrometry with a universal ATR sampling accessory (PerkinElmer Inc, Hopkinton, MA, USA) was used to analyse the functional groups of scaffolds at room temperature in the range 4000–500 cm$^{-1}$. The scaffolds were scanned 4 times per cycle at a resolution of 1 cm$^{-1}$. Perkin Elmer Spectrum software collected qualitative and quantitative data from the spectra.

### 2.6. Compression Test

The compression strength evaluations of scaffolds ($n = 3$), $2.4 \pm 0.2$ cm $\times$ $0.5 \pm 0.05$ cm, were performed with Lloyd LRX tensometer in 2.5 KN load cell compression mode. Before the compression test, all scaffolds were immersed in pH 7.4 PBS for 1 h. The scaffolds were compressed at a rate of 0.5 mm/min until 60% displacement. The Young's modulus values were calculated by the software.

### 2.7. Cytotoxicity Evaluation

Cell viability was assessed using the indirect contact method ($n = 6$). C2C12 cells, embryonic precursors of myocytes (muscle cells), were used for cell viability calculations. C2C12 cells were seeded at a density of $5 \times 10^3$ cells per well in 96 well plate and cultured until they reached over 80% confluence. Culture media consisted of Dulbecco's Modified Eagle's Medium (DMEM) high glucose media supplemented with 10% FBS, 1% penicillin-streptomycin and 1% L-Glutamine (Sigma Aldrich, Wicklow, Ireland). Scaffolds ($n = 3$) were immersed in medium and incubated for 24 h (Shown in Figure 1). Then, 100 μL elute medium replaced the old medium in the 96 well plate. Next, the cells were further incubated for 24 h and 48 h. At these endpoints, the cell viability was determined by measuring the activity of 3-(4,5-dimethylthiazol-2-yl)-2,5-diphenyltetrazolium bromide (MTT). The MTT assay measures the amount of formazan produced, which is proportional to the number of viable cells present. An amount of 100 μL (0.05 mg/mL) MTT solution was added in each well and incubated for 3.5 h in the dark. The MTT solution was replaced with 150 μL dimethyl sulfoxide (DMSO). Subsequently, the absorbance of scaffolds was measured at 570 nm (excitation/emission: 560/590 nm) using a Multilabel Counter Wallac 1420 (Perkin Elmer Inc, Hopkinton, MA, USA). Cell viability was calculated using the following equations:

$$\text{Viability}\% = (\text{sample} - \text{blank})/(\text{control} - \text{blank}) * 100 \tag{7}$$

blank: only media absorbance; sample: cell with scaffold absorbance; control: cell absorbance

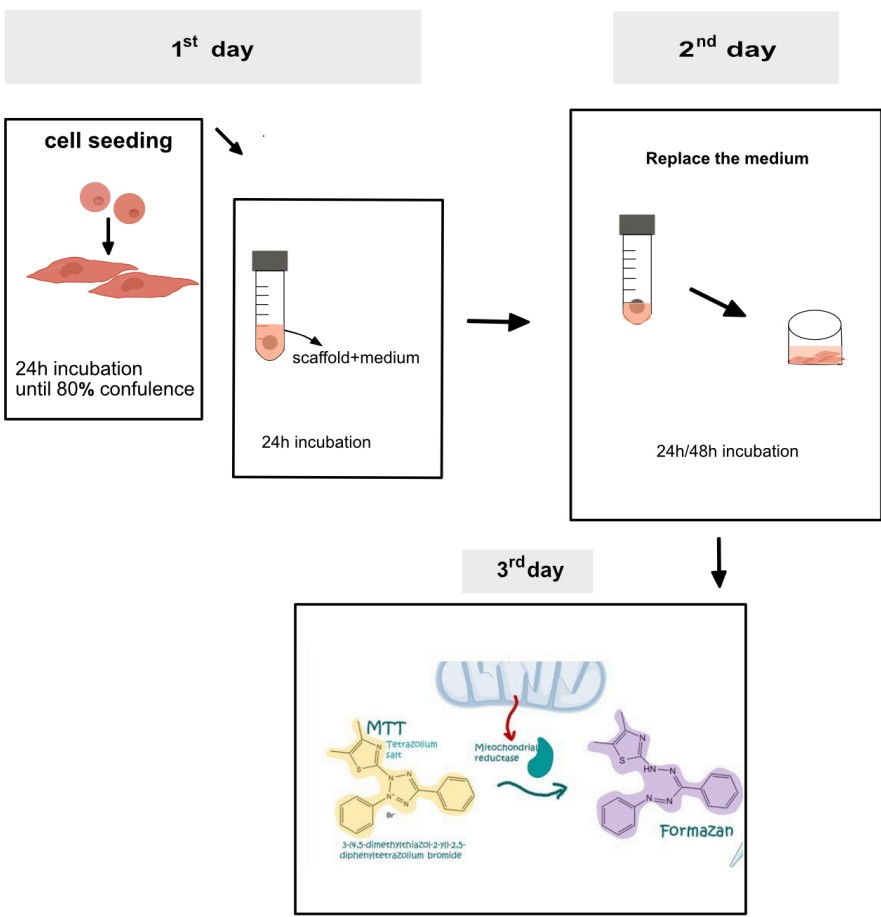

**Figure 1.** Flowchart of cell cytotoxicity evaluation test.

### 2.8. In Vitro Degradation Study

In the in vitro degradation study, the scaffolds ($n$ = 3) were immersed in simulated body fluid (SBF) at 37 °C for a duration of 8 weeks. SBF is an acellular solution with a chemical composition similar to that of the plasma. The preparation of SBF followed the Kokubo and Takadama protocol [28]. The scaffolds were weighed at $W_0$, $W_1$, $W_3$, $W_5$ and $W_7$. The weight of the scaffold in the first week was $W_0$. The degradation of the scaffold was calculated through weight loss measurements. The SBF was changed twice a week.

During incubation, a short term in vitro biomineralization study was evaluated simultaneously by removing samples at weeks 2 and 8 and drying the samples at 37 °C for 4–5 days. Analysis was performed by SEM/EDX to determine the deposition of apatite on the surface of the hydrogel-based composites, as described by Killion et al. (2013) [29].

### 2.9. Statistical Analysis

A statistical comparison of the swelling study, compression test and degradation test was performed using the Minitab statistics version 19 software. Following the assessment of normality of distribution and homogeneity of variance, treatments were compared using a one-way ANOVA with a Tukey's test to determine differences between individual batch groups, with a paired T-test used for cell viability. Differences were considered significant when $p \leq 0.05$.

### 3. Results and Discussion

In this work, a crosslinked scaffold was prepared by combining HAp/FAp with CS in the presence of PEG600DMA and a photoinitiator agent, benzophenone (BP), under UV light (shown in Figure 2). Visual inspection showed that the scaffolds were white due to

the homogeneous dispersion of the ceramic matrix (HAp/FAp) within the CS matrix. Post-curing, the scaffolds exhibited good flexibility and handleability. These results are similar to those reported by Zhang et al. (2013) [30] in which HAp particles were prominently visible on the surface of the scaffold [31].

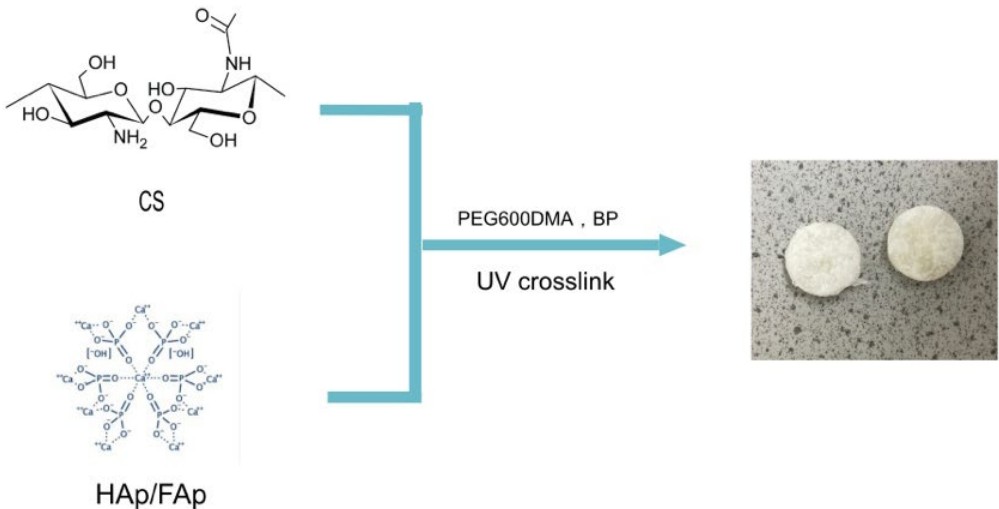

**Figure 2.** Schematic of the scaffold preparation.

### 3.1. SEM

Photomicrographs of the scaffolds were recorded using scanning electron microscopy in BSE mode (shown in Figure 3). The images indicated the presence of materials with different densities, while EDX showed the heterogenous distribution of Ca and P throughout the matrix, which was indicated by the presence of HAp/FAp, which may be beneficial in cell adhesion and the mechanical strength properties of the scaffold [32]. Fluorine ions were not detected in EDX, which was likely due to the low level of fluorine substitution in HAp. With the increase of FAp in the scaffold, the Ca/P ratio varied from 2.1 to 1. The reduction in the Ca/P ratio could be due to unreacted P from $H_3PO_4$ used in the reaction for the preparation of FAp, indicating that additional washing steps may be necessary in the preparation of FAp. Phosphoric acid produces hydrogen ions, which can bond with fluorine ions in sodium fluoride through hydrolysis. Thus, it may produce additional P as a by-product [33,34]. The main consistuent of the bone mineralisation phase is hydroxyapatite, which includes 39.9% calcium and 18.5% phosphorus with a Ca/P ratio of 2.16 [35]. Compared to Azaman et al.'s (2022) [36] work, the Ca/P ratio is lower, though the FAp scaffolds had a similar ratio.

The essential requirement for the bone scaffold to bond to living bone is to form an apatite layer on the surface of the scaffold [37]. The biomineralization properties were determined by incubating the scaffold in vitro for weeks 2 and 8. At weeks 0, 2 and 8, the samples were removed and assessed by SEM/EDX. SEM images in Figure 3 of the scaffold before and after incubation with SBF showed that a mineral-like layer was formed on the scaffolds' surface. The structures of globular crystals were shaped like cauliflowers [38,39]. The EDX results were investigated to obtain the elemental composition of the mineral layer. The Ca/P ratios of the scaffolds had a variety of changes after incubation with SBF because the scaffolds were uptaking the $Ca^{2+}$ and $PO_4^{3-}$ ions from the SBF solution (shown in Table 2) [40,41]. In the present work, it was found that only the FAp scaffold exhibited poor biomineralization in vitro compared to the other scaffolds. This could be attributed to the porosity of the ceramic and is in agreement with the fact that the mineral layer formed faster with the HAp scaffold, as reported by Borkowski et al. (2021) [42].

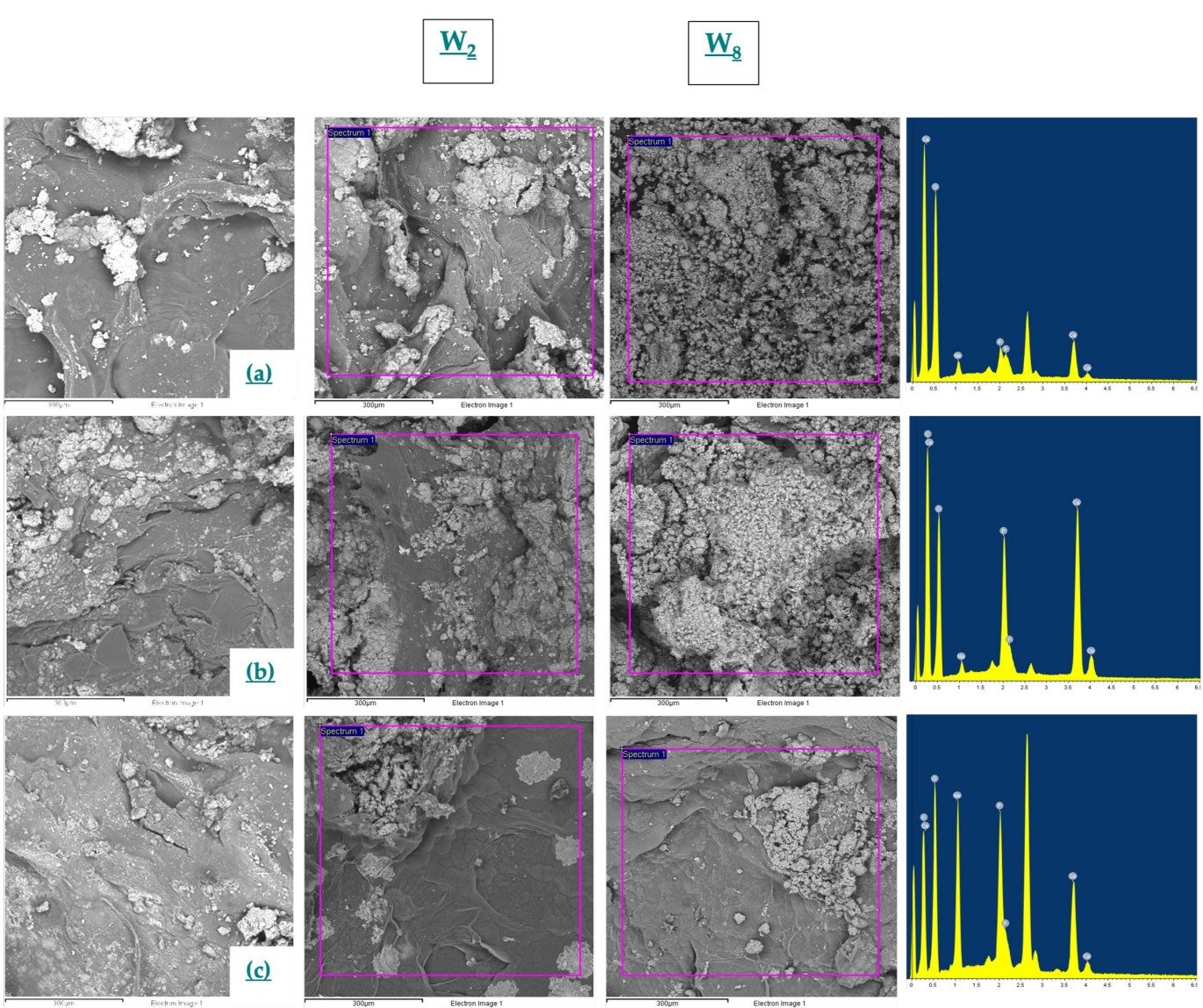

**Figure 3.** SEM view and EDX mapping of 12.5 $CS_H/HAp_{1.5}/FAp_0$ (**a**), 12.5 $CS_H/HAp_{0.75}/FAp_{0.75}$ (**b**), 12.5 $CS_H/HAp_0/FAp_{1.5}$ (**c**) and after soaking in simulated body fluid at weeks 2 and 8.

**Table 2.** Ca/P ratios of bone-like apatite layers released on the scaffolds.

| Atomic Percent (%) | 12.5 $CS_H/HAp_{1.5}/FAp_0$ | | | 12.5 $CS_H/HAp_{0.75}/FAp_{0.75}$ | | | 12.5 $CS_H/HAp_0/FAp_{1.5}$ | | |
|---|---|---|---|---|---|---|---|---|---|
| | **Ca** | **P** | **Ca/P ratio** | **Ca** | **P** | **Ca/P Ratio** | **Ca** | **P** | **Ca/P Ratio** |
| $W_0$ | 9.821 | 3.39 | 2.89 | 9.9 | 4.99 | 1.98 | 5.82 | 6.18 | 0.94 |
| $W_2$ | 24.56 | 10.29 | 2.38 | 15.79 | 7.49 | 2.1 | 8.82 | 3.54 | 2.94 |
| $W_8$ | 29.53 | 13.25 | 2.22 | 30.88 | 13.15 | 2.34 | 9.87 | 3.6 | 2.7 |

### 3.2. Swelling Study

Swelling studies were conducted to investigate the effect of varying concentrations of HAp and FAp on the gel fractions (GFs) of scaffolds (shown in Figure 4). The GFs results indicate that the incorporation of HAp (82.37 ± 5.6%) resulted in a higher GF compared to the equivalent Fap-containing scaffold ($p$ = 0.037). There was no obvious difference in the scaffolds that contained both Hap and Fap (80.06 ± 7.6%) and Fap (79.58 ± 2.6%), respectively. The scaffolds with higher ratios of FAp had the lowest number of GFs, suggesting that the presence of FAp may have inhibited crosslinking. The FAp may have

interrupted the crosslinks between monomeric chains, which prevented the polymeric chains from forming covalent links to the network, thereby allowing them to diffuse out of the system during swelling [38,43]. Nonetheless, the scaffold containing FAp did yield GFs of ca. 80% and, as such, its presence did not completely interrupt the network connectivity, and in all scaffolds, crosslinking occurred as, alternatively, the scaffolds would have completely dissolved [29].

To assess the effect of varying volumes of AA on the GFs of the scaffolds, 12.5 $CS_H/HAp_0/FAp_{1.5}$, 20 $CS_H/HAp_0/FAp_{1.5}$ and 30 $CS_H/HAp_0/FAp_{1.5}$ were calculated as 79.58 ± 2.6%, 74.37 ± 5.08% and 72.36 ± 8.39%, respectively (Figure 4). GFs of 72.36–79.58% indicated that a high portion of the polymers were crosslinked. Based on the observed data, the scaffolds with lower volumes of AA had higher GFs ($p = 0.002$), indicating a higher degree of crosslinking and network formation [36].

Bone is made up of 15–25% water by volume [44]. The equilibrium water content (EWC) of the scaffolds produced using varying volumes of AA ranged between 91.3 ± 0.5% to 95.6 ± 0.4%. However, the EWC of the scaffolds with varying HAp/FAp ratios ranged from 62 ± 0.6% for the Hap-containing scaffold, which increased to between 92.5 ± 0.6 and 91.4 ± 0.5% with the incorporation of FAp ($p < 0.05$). Azaman et al. [36] studying FAp scaffolds, found similar results. In this work, it is hypothesised that the fluorine ions attracted and held higher volumes of water in the scaffold and may have led to lower crosslink levels and porosity of the scaffold and increased the scaffolds' water absorption capacities [36,45].

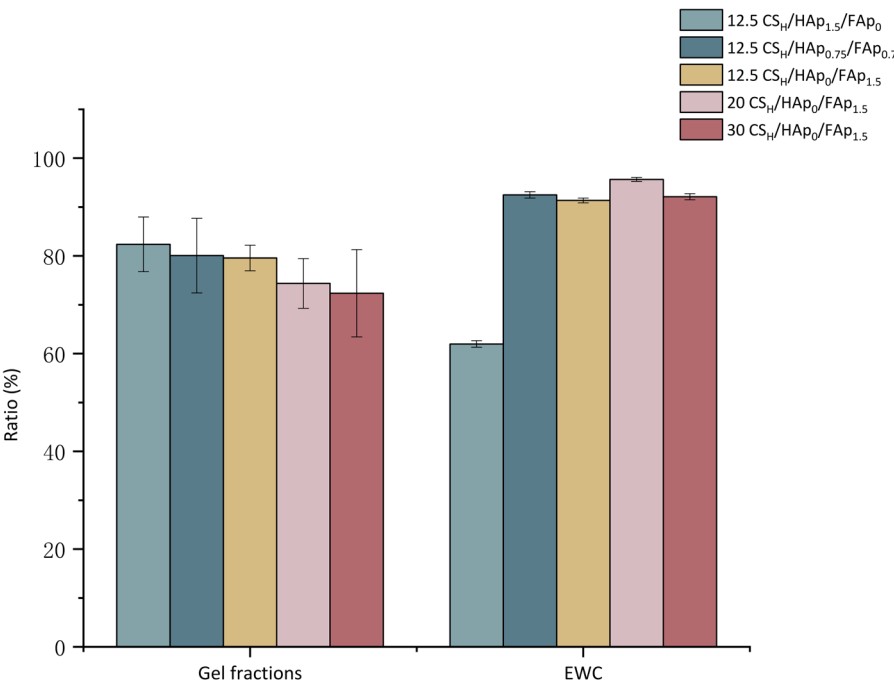

**Figure 4.** Gel fractions and EWC of scaffolds.

The water uptake ($W_U$) and percentage swelling data followed the same trends for all scaffolds tested. On examination of the $W_U$ and percentage swelling data of scaffolds with varying HAp/FAp ratios, it was indicated that HAp may have retarded swelling ($p < 0.05$). The significant change in the percentage swelling of the HAp scaffold could be attributed to the hydrophobic nature of Hap, which reduced the hydrophilicity of chitosan by binding to hydrophilic $NH_2$ and OH [46]. Azaman et al. [36] reported that HAp more easily bound the polar groups of CS, thereby reducing electrostatic repulsion between the polymeric chains and preventing high levels of swelling and degradation. However, the incorporation of FAp caused the scaffold to obtain a higher percentage of swelling regardless of the presence of HAp. Thus, the appropriate combination of FAp may produce

scaffolds with optimal porosity, which benefits the scaffolds' degradation rate and matches new bone formation rates [37,47].

In the analysis of the scaffolds prepared with varying volumes of AA, it was found that scaffolds containing 20 mL of AA/1.5 g of CS had the highest $W_u$ and percentage swelling ($p < 0.05$); it was almost double that of the $12.5/30$ $CS_H/HAp_0/FAp_{1.5}$ scaffolds tested (shown in Figure 5). The 20 $CS/HAp_0/FAp_{1.5}$ scaffold may maintain more polymer molecules within the system, allowing them to swell and entrap water. The data suggested that the properties of the scaffold may be altered due to variations in the volume of AA. The protonation amino group of CS increased with an increasing AA volume, which resulted in the chitosan matrix absorbing more water because of stronger hydrogen bonding [17]. Conversely, the other scaffolds had a higher dissolution rate, thereby reducing the swelling data recorded.

Swelling behaviour affects the scaffolds' 3D structure and mechanical strength. After the scaffold becomes swollen, the pore size of the scaffold will increase, which can enhance cell attachment and migration but can adversely affect the mechanical properties of the scaffold [48].

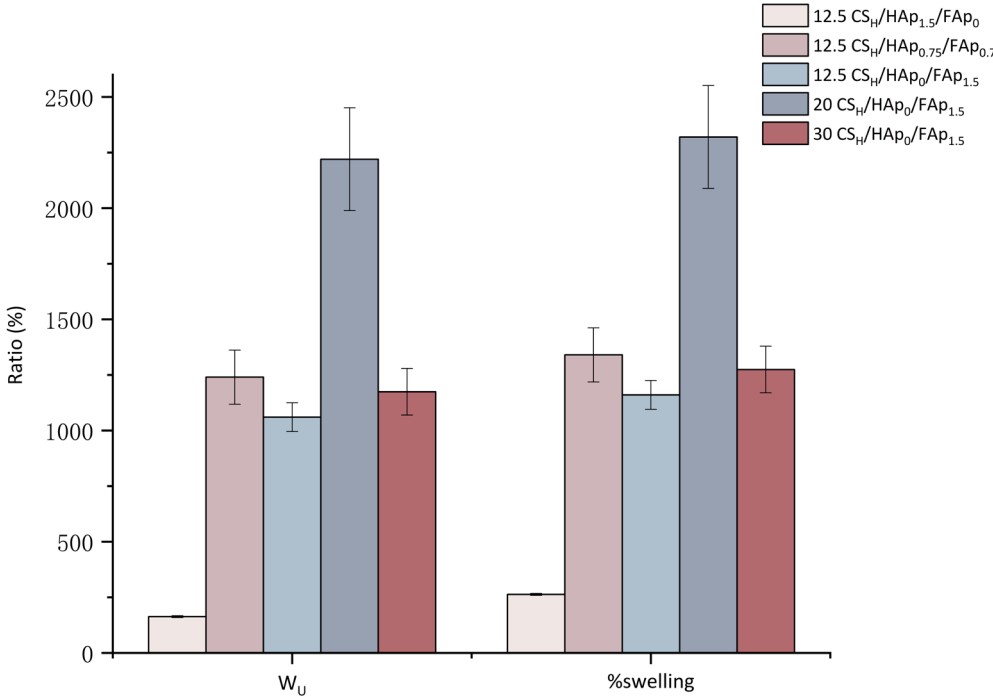

**Figure 5.** $W_U$ and percentage swelling of scaffolds.

*3.3. FTIR*

The characteristic bonds of CS were observed in the following spectra: 3500–3000 cm$^{-1}$ was the overlap of alcohol (O-H) bonds of primary and secondary amides, 1200–1000 cm$^{-1}$ indicated C-N stretched aliphatic amines, and 800 cm$^{-1}$ indicated the N-H bond waves. The peaks at 1647 cm$^{-1}$ and 1606 cm$^{-1}$ were the characteristic peaks of the C=O bond in the N-Acetyl group of the CS. The sharp peak at 1409 cm$^{-1}$ was assigned to the CH$_3$ symmetric deformation mode. The peaks of 1074 cm$^{-1}$ and 1031 cm$^{-1}$ corresponded to the C-O stretching vibration $\nu$ (C-O-C) [37] (shown in Figure 6).

The characteristic absorption peaks of the Hap and FAp were similar due to their similar chemical structures, and were: the stretching mode of the OH$^-$ group appeared at 3500 and 1651 cm$^{-1}$; the peak corresponding to the internal modes of the PO$_4^{3-}$ group appeared at $\nu_4$ (600 and 570 cm$^{-1}$), $\nu_3$ (1085 and 1033 cm$^{-1}$) and $\nu_1$ (960 cm$^{-1}$) [49]. In addition, the presence of a small band at ca. 746 cm$^{-1}$ in all spectra of FAp ceramics may

indicate the presence of hydrogen bonding of the hydroxide group with the fluorine ion group of FAp [42].

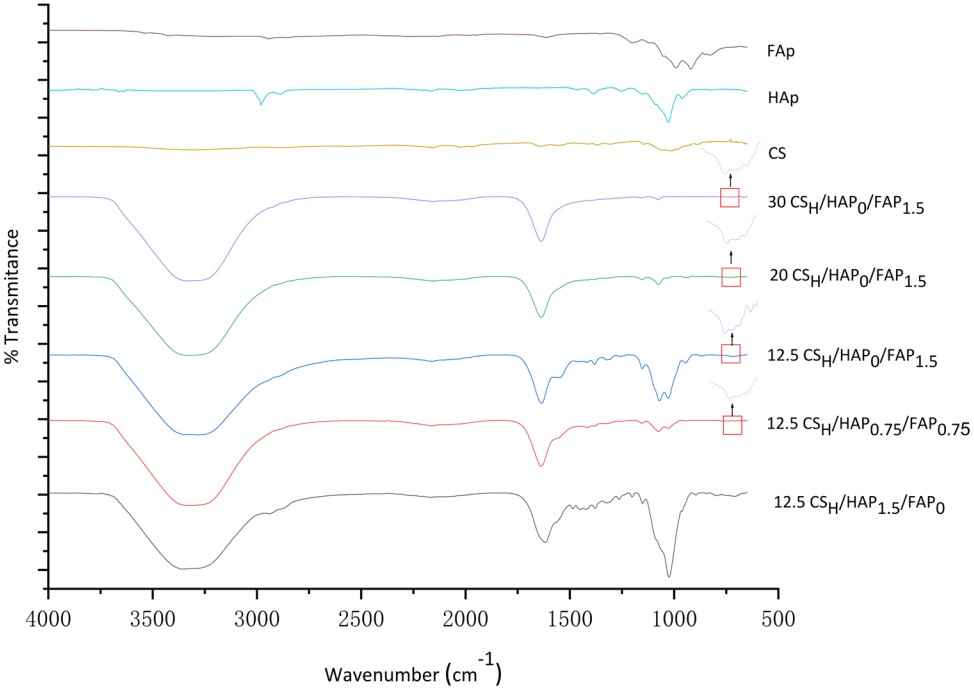

**Figure 6.** FTIR spectra of different scaffold formulations indicated the presence of characteristic peaks of CS and HAp/FAp following UV crosslinking reaction.

During scaffold preparation, the CS powder was dissolved in different volumes of AA. In this process, the vibrational frequency of $NH_2$ was weaker because of the protonated amino groups ($NH_3^+$) generated with the increasing volume of AA [50]. The 12.5 $CS_H/HAp_0/FAp_{1.5}$ scaffold had more peaks in the frequency range of 1623–1069 $cm^{-1}$ compared to those of 20 $CS_H/HAp_0/FAp_{1.5}$ and 30 $CS_H/HAp_0/FAp_{1.5}$ (shown in Figure 6). Specifically, the absorption bands at 1544 $cm^{-1}$, 1383 $cm^{-1}$ and 1313 $cm^{-1}$ correspond to N-H shifts, the symmetric deformation of acetyl groups and pyranose ring vibrations (CS and FAp), which disappeared at higher concentrations of AA [37]. The 1635 $cm^{-1}$ band was likely the result of the decrease in the stretching frequency due to proper dispersion of CS and HAp/FAp [51]. Different volumes of acetic acid may result in variations in the FTIR spectra. It may cause peak positions and intensity changes in FTIR spectra because of amine group protonation. After the CS was dissolved in different volumes of AA, the CS molecular conformation and hydrogen bonding may have interacted at different levels [50].

Furthermore, the hydroxyl groups of 12.5 $CS_H/Hap_{0.75}/Fap_{0.75}$ and 12.5 $CS_H/Hap_0/Fap_{1.5}$ were shifted to the right when compared to 12.5 $CS_H/Hap_{1.5}/Fap_0$, which indicates that hydrogen bonding occurred between the fluorine ion and hydroxyl group in the ceramic compounds [52]. The peak of 702 $cm^{-1}$ was attributed to the proper dispersion of HAp in the CS polymer and HAp ceramic matrix. Borkowski et al. [21] reported bands at 963 $cm^{-1}$, 1025–1032 $cm^{-1}$ and 1088 $cm^{-1}$ for the phosphate group in HAp/FAp [42]. The scaffolds' FTIR were similar due to HAp and FAp having similar chemical structures. However, after the addition of FAp, the intensity of the 3300 $cm^{-1}$ peak appeared to have a reduced intensity because of the replacement of the hydroxyl groups with fluorine ions in the FAp ceramic [53].

The reduced stretching frequency of CS in CS/HAp/FAp indicated that there may have been some intermolecular and coordinating chemical interactions between CS and calcium ions of HAp/FAp particles in the scaffold [21]. These chemical interactions make the composite scaffold stiffer and more stable [54].

### 3.4. Compression Test

Based on experimental data, $30\,CS_H/HAp_0/FAp_{1.5}$ had the lowest mechanical strength among scaffolds with varying volumes of AA (shown in Figure 7) (*p* = 0.006). The 30 $CS_H/HAp_0/FAp_{1.5}$ scaffold had a high $W_u$ and low GFs, which may explain the relationship between swelling and mechanical strength. The lower GFs led to the higher porosity of scaffolds, which resulted in the scaffold's mechanical strength being reduced [55,56].

After adding the ceramic to the scaffold, the mechanical strength increased (shown in Figure 7). Mechanical testing data of scaffolds with varying ratios of HAp and FAp, indicated that scaffolds containing HAp recorded the highest modulus (*p* = 0.002). In previous work, it was found that the polarity of HAp increased the crosslinking of the UV curing reaction, which led to an increase in the mechanical properties. Nevertheless, the observed values increased compared to previous work within our laboratory [57]. FAp particles have a lower porosity than HAp, and the compressive strength decreases with increasing porosity [58]. It was observed that HAp scaffolds had better mechanical behaviour than FAp scaffolds, indicating that FAp interacted with the CS polymeric chains of the scaffold in 12.5 mL AA.

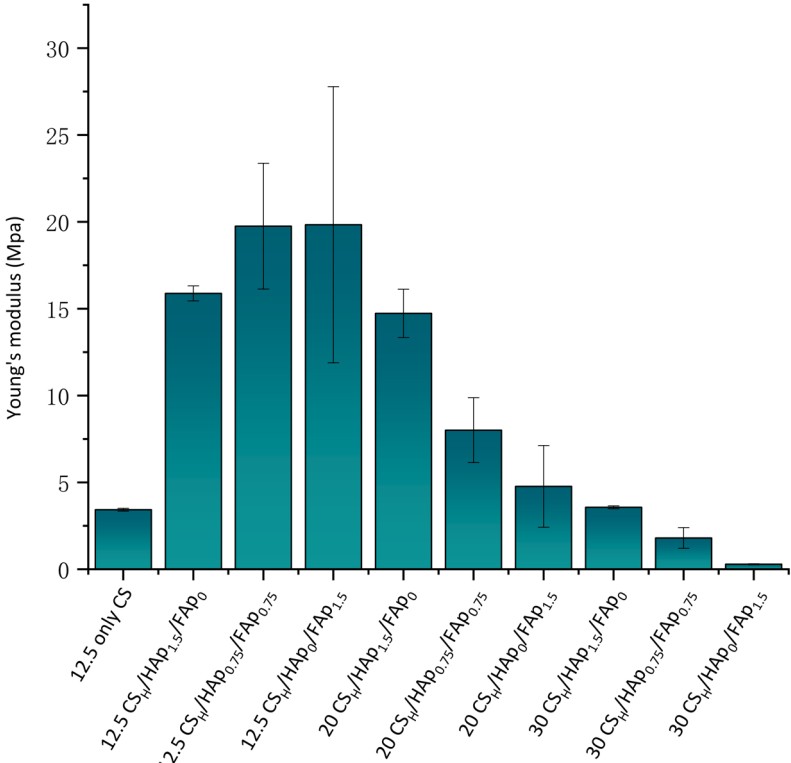

**Figure 7.** Mechanical properties of scaffolds.

The strength and modulus of the elasticity/stiffness of the bone scaffold material are important. The scaffold needs to withstand early frontal biological forces in the new bone formation process [55,59]. In addition, at the bone–scaffold contact surface, mechanical instability and micro-movements of the scaffold may produce wear debris. This wear debris can ultimately result in inadequate osseointegration or other adverse biological reactions [60].

### 3.5. Cytotoxicity Evaluation

In vitro cytotoxicity evaluation is essential for bone scaffold biocompatibility properties [16,21]. Mouse mesenchymal precursor cells of C2C12 line viability on CS/HAp/FAp scaffolds were evaluated by the MTT assay for 24 and 48 h. In this experiment, C2C12 cells

were seeded with different scaffold elute media to study the effect of the volume of acetic acid and the ratio of HAp/FAp on cell viability.

The results showed that scaffold elute medium caused a varied reduction in the viability of C2C12 cells in the range of 31.6–76.4% in comparison with untreated cells. Hence, the cell viability significantly decreased when incubated with the 12.5 $CS_H$/$HAp_0$/$FAp_{1.5}$ scaffold ($31 \pm 8.8\%$), while the cell viability increased with an increasing AA volume in the scaffold (30 $CS_H$/$HAp_0$/$FAp_{1.5}$ is $66.17 \pm 11.9\%$) (shown in Figure 8). The high concentration of fluorine ion/unreacted compounds from the preparation of FAp, appeared to be harmful to the cell, even though the exact mechanism of fluoride toxicity is unclear [61].

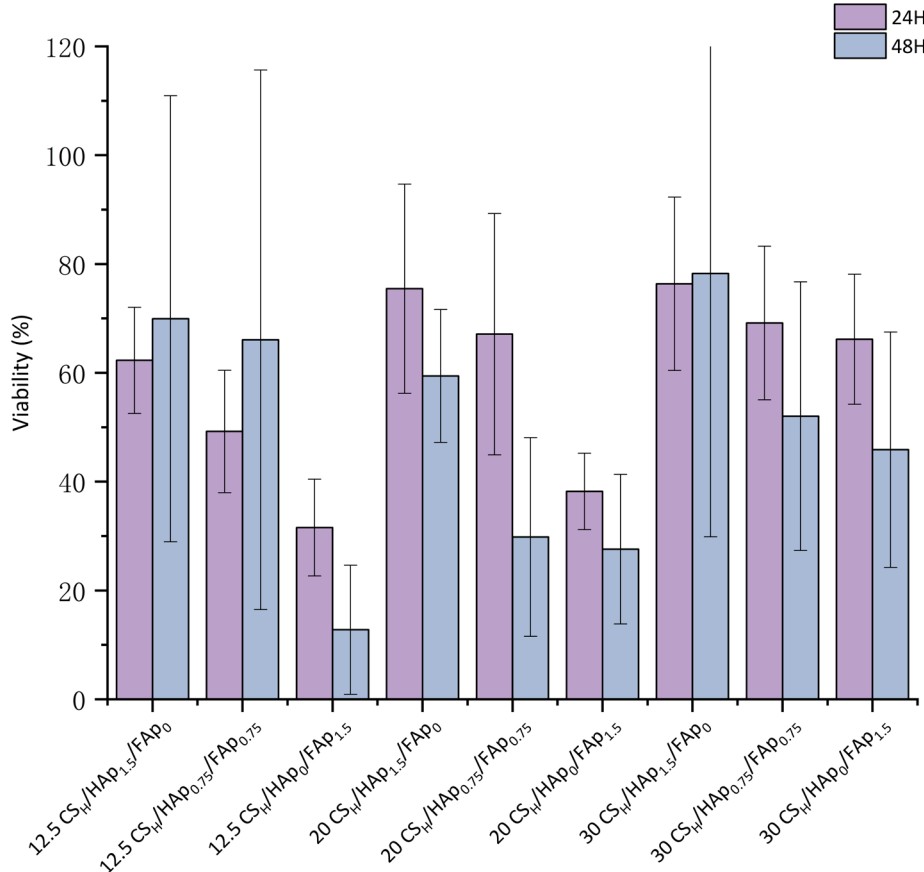

**Figure 8.** Cytotoxicity analyses of C2C12 cells on different scaffold formulations.

The treatment of C2C12 cells with the 20 $CS_H$/$HAp_{1.5}$/$FAp_0$ and 30 $CS_H$/$HAp_{1.5}$/$FAp_0$ scaffold extracts did not have a substantial effect on cell viability ($75.46 \pm 19.2\%$ and $76.36 \pm 15.93\%$). According to ISO 10993-5, 100% extracts of biological material that do not reduce cell viability by more than 30% (cell viability > 70%) should be considered non-toxic [62]. Furthermore, the performance in in vitro cell studies revealed good cell viability of the 20 $CS_H$/$HAp_{1.5}$/$FAp_0$ ($75.46 \pm 19.2\%$), 30 $CS_H$/$HAp_{1.5}$/$FAp_0$ ($76.36 \pm 15.39\%$) and 30 $CS_H$/$HAp_{0.75}$/$FAp_{0.75}$ scaffolds ($69.16 \pm 14.12\%$) ($p > 0.05$, no significant).

*3.6. In Vitro Degradation Study*

The scaffold degradation rate is a key element in the bone regeneration process. The scaffolds' degradation rates were evaluated with weight loss at weeks 0, 1, 3, 5 and 7 after immersion in an SBF solution. The scaffolds' ideal degradation rates should match the new bone growth rate, which is typically 6 to 8 weeks [63]. Finally, the scaffolds should be gradually integrated with the bone tissue after degradation.

In the initial four weeks, the HAp scaffolds degraded slowly compared with the FAp scaffolds ($p = 0.023$). In the comparison of the different volumes of acetic acid in the

scaffolds, the 20 mL AA scaffold degraded the slowest in four weeks, which collates with the 20 mL scaffold having the highest number of GFs. The sudden weight loss of the FAp scaffold at the first and fourth weeks was possibly owing to the lower GFs or the dissolution of the unreacted components from the preparation of FAp.

Furthermore, the scaffolds exhibited a continuous decrease in weight loss after soaking in SBF for 8 weeks (shown in Figure 9). The 30 $CS_H/HAp_{0.75}/FAp_{0.75}$ scaffold, at 11.8%, had the lowest rate, and the 20 $CS_H/HAp_0/FAp_{1.5}$ scaffold had the highest rate of degradation at 72.6% ($p < 0.05$). Comparing the weight loss of the scaffolds, the degradation of the FAp scaffold was faster than the other scaffolds for all the tested time points. This finding may be attributed to FAp having a higher percentage of swelling, resulting in the scaffold more easily breaking down in solution. After the FAp scaffold was swollen, it offered more surface to interact with the SBF in vitro [37]. The FAp scaffold stability was lower than that of the HAP scaffold, which indicated a higher bioresorbability of the FAp scaffold. The good solubility of the FAp scaffolds is a desirable characteristic considering their BTE application as a bone scaffold material [47].

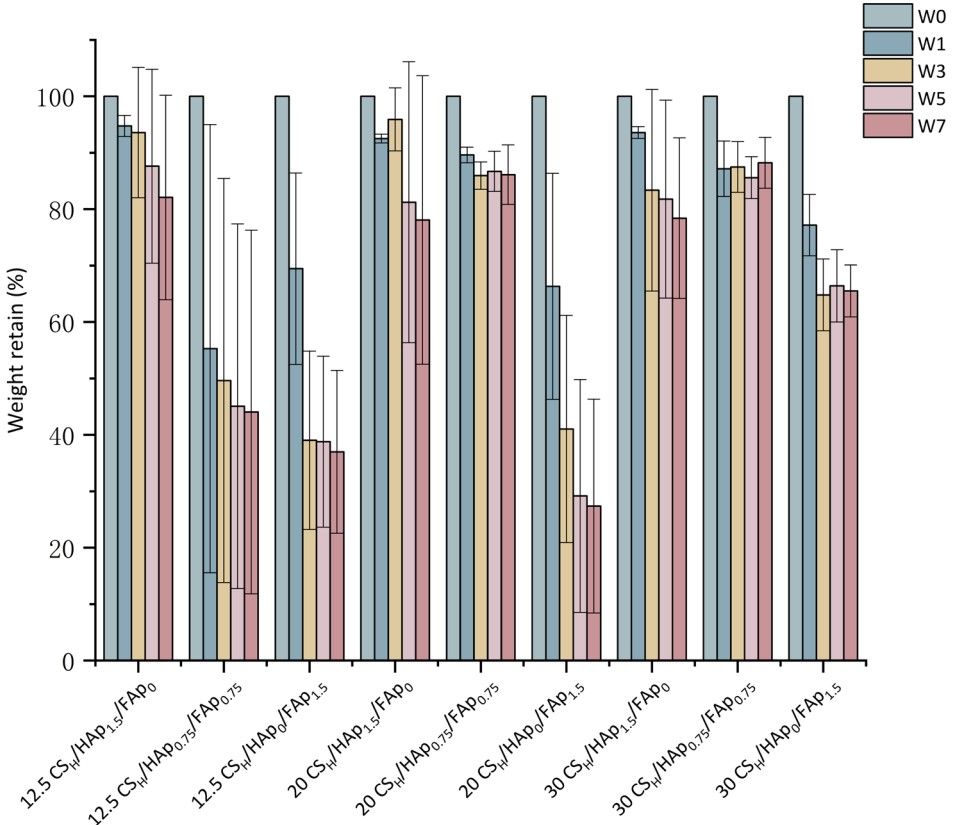

**Figure 9.** Scaffold weight loss in $W_0$, $W_1$, $W_3$, $W_5$ and $W_7$ time points.

## 4. Conclusions

This study investigated the preparation of a novel bone scaffold that exhibits good mechanical strength, biocompatibility and biodegradability. The scaffold was made through a combination of chitosan and bioactive ceramics, which were crosslinked through a photopolymerisation reaction under UV. The effects of different volumes of AA and ratios of HAp/FAp were discussed through the crosslinking, biocompatibility and mechanical properties results.

FTIR spectra identified the presence of CS, FAp and HAp in the scaffolds and intermolecular bonding between the CS and calcium ions of the HAp/FAp particles. The in vitro degradation results showed that the scaffolds degraded gradually over 8 weeks, which was owing to the degradation of hydrophilic polymers (CS) present in the scaffold. The

MTT assay results indicated that some scaffold formulations were non-toxic to cells. FAp did exhibit some cytotoxicity when used; however, it should be noted that FAp synthesis process, and as such, a rigorous washing of the FAp will likely result in a reduction of the toxicity level recorded [64]. The results indicated that the scaffold formulations were nevertheless biocompatible and may have the potential for use as bone scaffolds with further optimisation in the future.

Overall, these results suggest that the swelling behaviour and mechanical strength of scaffolds can be manipulated by adjusting the volume of AA or the ratio of HAp and FAp. These findings may have implications for the development of scaffolds in tissue engineering applications, where the mechanical properties and water absorption capacity of the scaffold are important considerations.

**5. Future Work**

This work has proven that the proposed scaffold has good swelling behaviour, mechanical strength and biocompatibility. The next steps in scaffold development will be to improve the cytocompatibility and enhance the osteoconductivity and osteoinductivity of the scaffolds.

**Author Contributions:** Conceptualisation, D.M.D., M.B.F. and Z.C.; methodology, D.M.D., M.B.F., Z.C. and K.Z.; investigation, K.Z., F.A.A. and M.B.F.; data curation, D.M.D., M.B.F., Z.C. and K.Z.; writing—original draft preparation, K.Z.; writing—review and editing, K.Z. and D.D; supervision, D.M.D., Z.C. and M.B.F.; project administration, D.M.D.; funding acquisition, D.M.D. All authors have read and agreed to the published version of the manuscript.

**Funding:** Funding for this work was provided by the Technological University of The Shannon through the President Seed Fund and the Enterprise Ireland Commercialisation Fund grant number CF20160600.

**Data Availability Statement:** Not applicable.

**Conflicts of Interest:** The authors declare no conflict of interest.

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
