# Peer review of "Bone Tissue Engineering Scaffold Optimisation through Modification of Chitosan/Ceramic Composition"

_2673-6209, doi:10.3390/macromol3020021_

Round 1
Reviewer 1 Report
The aim of this study is to examine the potential use of CS/HAp/FAp scaffolds in place of CS/HAp scaffolds for the preparation of the BTE scaffold which is a biocompatible, mechanically robust scaffold with stability suitable for bone healing. This research is under the scope of this journal; the topic is relevant for readers, and this research deals with potentially significant knowledge of the field. And It will be important for Bone tissue engineering knowledge. The topic is relevant for readers and this study deals with potentially significant expertise in the field and opens a new way for future studies.
However, there are some aspects which are needed to be improved in the manuscript:
- Correct typos in all manuscripts.
(Abstract)
- In the results, is important to show more information, and add some of the p-values.
(Keywords)
- Please add keywords, and order the keywords / Mesh terms alphabetically
- Remove the author's instructions from all manuscript.
(Introduction)
What is the importance of this study? What is the gap in this field of literature?
- Lines 28 - “Bone is an indispensable and multifunctional organ” add a reference also the tissue bone is a dynamic tissue, please read also this article https://doi.org/10.3390/molecules26051339 for support the importance of the bone tissue graft. Critical size defect on the bone tissue? The scaffolds are crucial for bone tissue regeneration in critical-size defects. It seems crucial for biodegradation For this reason the authors need to support the necessity of using Scaffolds in “Critical size defect” by example the animal study.
- It was investigated in an animal study the usage of chitosan scaffolds with very good biocompatibility for dental regeneration (Dentistry). investigated in an animal study the usage of this lyophilized hydrogel Chitosan scaffolds for dental regeneration. This was added inside the root canal dentine walls to see recovery of dental tissues. However, this procedure has seen an increase periodontal tissue inside the Root canal with regeneration as bone tissue!
- Please, add more information about the importance of bone grafts. the space provides a limitation for some bone graft material.
- Regeneration bone defects with scaffolds of the pores or the space provision versus compacts materials, Can you describe the importance of the porous or the space provision on new material versus stabilization of the bone. please read this (Palma, P. J., (2010), New formulations for space provision and bone regeneration. Biodental Eng. I, 1, 71-76. WOS:000282776500012; SBN 978-0-415-57394-8) reported the influence of different formulations of bone grafts in providing an adequate scaffold, thus emphasising the importance of the type of carrier in the three-dimensional distribution of particles and also space provision in new bone formation. Please described, did the size of your particles allow the maintenance of space for bone formation?
- Also described the differences between bone fillers and bone substitutes. These synthetic grafts will work as bone fillers or bone substitutes?
- You do not think this study is included in the others already done? Which results are comparable with other studies? What has this study been new?
- Identified the aim and null hypothesis at the end of the introduction
(Materials and Methods) .
- remove lines 81 to 87.
- When mentioning materials or devices: please mention the manufacturer and city/ country.
- How many operators performed and evaluated the experiments? How many observers did the scoring? What is the interobserver agreement?
- How was the sample calculated for MTT? Did the authors perform a power analysis to evaluate whether this group sample size was appropriate? Please, add this information experimental design.
- Improve the resolution quality of all figures and graphs (and a presentation). The font/language in the figure/caption is different from the text. Please, standardise the size and the font in the figures and charts with the font of the manuscript.
(Discussion)
- Need to add discussion on this type of article.
- Please, identified what was the strength(s) and add more limitations of this study. And also, implications for future perspectives.
References
- And when you had in the text the “authors et al.” references should come immediately afterwards, not at the final of the sentence.
Author Response
Dear Editor,
On behalf of my co-authors, we thank you very much for giving us an opportunity to revise our manu, we appreciate the editor and reviewers very much for their positive and constructive comments and suggestions on our manu entitled one tissue engineering scaffold optimization through modification of chitosan/ceramic composition.
We have studied the reviewer’s comments carefully and have made revisions which are marked in red in the paper. We have tried our best to revise our manu according to the comments. Attached please find the revised version, which we would like to submit for your kind consideration. We would like to express our great appreciation to you and the reviewers for your comments on our paper. Looking forward to hearing from you.Thank you and best regards.
Kind regards
Keran Zhou

Reviewer 2 Report
Background
Introduction addresses the key issues of relevance, specifically the hypothesis concerning chitosan/HA/FAP scaffolds, the potential benefits of chitosan, the potential benefits of HA and FAP. The authors make adequate use of the literature to place their work in appropriate scientific context.
Methodology is adequately described, in spite of the extraordinary error in incorporating author instructions. I concur with the suitability of swelling, compressive strength, cytotoxicity and in vitro degradation for the experimental characterisation methods. The cell model and methodology are appropriate for the purpose.
Findings
The data presented shows scientific findings, adequately described, and an adequate quantity of data for a scientific paper. However, given that both HA and FAP are being used, each with differing anticipated in vivo tissue response as discussed in the introduction, one wonders why cytology and biodegradation were the only issues tested in vitro. Some SBF studies, with the use of some pure chitosan control specimens, could have been easily conducted and could have provided significant information regarding potential in-vivo response. Even better, with the use of some pure chitosan control specimens, some actual cell studies could have been done using osteoblast or even just fibroblasts to establish some in-vitro guidance on the differences in HA and FAP effects, and the bioactivity benefits of the apatite additions. Without such testing, what was the point of using two fundamentally different types of bioactive apatite? All that has been established is cytotoxicity and swelling effects, which shed very little light on the usefulness of this material in its stated purpose – synthetic bone grafts.
Presentation
The written English of this manuscript is of a very poor standard. It needs a major copy edit by a native English speaker.
The Materials and Methods section is prefaced with author instructions. The Results section is also prefaced with author instructions. In all my years of reviewing I have never encountered such a cardinal error in manuscript presentation as this, and it makes the reviewer question if the correct version of the manuscript was submitted. Is this submitted manuscript an early draft perhaps? This is a manuscript with 5 authors, and yet not one of them noticed this extremely obvious error. It does not inspire confidence in the professionalism of the authors.
I did not waste further time looking for other presentation errors, but I imagine there are probably others. The authors need to thoroughly proof their manuscript prior to resubmission.
The authors have failed to reference figure 3 in the text, another significant error in presentation.
Author Response

(The authors gave the same response as above.)

Reviewer 3 Report
Dear Authors,
The article is good. And in principle, I have nothing to comment on. Substantively everything is fine. The only thing I ask you to do is to improve the SEM pictures - because they are of very poor quality. And above all, they are too small. Another problem is the lack of EDX spectra. Please correct this and the article will be ok.
Regards
Reviewer
Author Response

(The authors gave the same response as above.)

Round 2
Reviewer 1 Report
The authors improved the article with the reviewer`s comments.
Reviewer 2 Report
My concerns raised in review 1 have been adequately addressed. Manuscript is acceptable for publication.
Needs a final copy edit by a native English speaker.